# Flexural Strength and Stiffness of Donut-Type Voided Slab

**Joo-Hong Chung [1], Hyung-Suk Jung [2] and Hyun-Ki Choi [2,\*]**

[1] Division of Architectural Engineering, Daejin University, 1007, Hoguk-ro, Pocheon-si 11159, Korea; scarletmoon@daejin.ac.kr

[2] Department Fire and Disaster Prevention Engineering, Kyungnam University, Changwon 51767, Korea; junghs@kyungnam.ac.kr

\* Correspondence: chk7796@kyungnam.ac.kr

**Abstract:** The voided slab system has been known as an effective technique to replace a heavy reinforced concrete slab system without the decrease in flexural strength. However, according to the previous studies, the flexural capacities such as flexural strength, stiffness and ductility of the voided slabs were practically lower than that of the solid slabs depending on the void shapes and details. Therefore, in this study, an analytical and experimental study were conducted to derive the optimal void shape and details focused on the flexural capacities of voided slabs. Based on a finite element (FE) analysis, a donut-type void shaper, which was a hexahedron with rounded edges and a hole penetrating the center, was suggested as the optimal shape in voided slabs, and an experimental study was conducted to verify flexural capacities of the donut-type voided slab. The flexural strength, stiffness and deflection of the donut-type voided slab were investigated by void shape and fixing method of void shaper as variables. The ductility of voided slab was also evaluated, because ductility is as important as strength for the safe design of slab member. The test results showed that the flexural strength of the donut-type voided slabs was equivalent to 98% and 105% that of the solid RC specimen, and the donut-type voided slab specimens had enough ductility for the flexural member. The stiffness of the donut-type voided slab was decreased about 8~9% compared with the solid slab, but it was improved up to 7% compared to the non-donut-type voided slab. Based on test results, the flexural design method of the donut-type voided slab associated with the void shape and fixing device of void shaper was suggested, and it was confirmed that the donut-type voided slab is one of the efficient alternatives to replace heavy flat plate slabs.

**Keywords:** donut-type voided slab; void shape; flexural capacity; finite element analysis; experimental investigation

## 1. Introduction

### 1.1. Research Scope and Objectives

The increase of the length of span in buildings have brought about the increase in slab thickness in an attempt to mitigate slab deflection, noise and vibration. However, the application of thicker slabs weighing more causes the increase in the size of vertical members such as columns, walls, and bases, which results in the rise in the overall weight of a building and the amount of material. The increase in building weight and the amount of material is a negative factor in construction because it deteriorates the building's economic efficiency and increases seismic load.

In the early 20th century, voided slab systems were developed using a segmented void shaper such as spherical or oval plastic balls for two-way slab applications, and the segmented void shaper was expected to eliminate the slab's directivity and reduce its weight while maintaining its flexural capacity [1]. The main idea of the voided slab system is that lightweight void shapers are placed between the top and the bottom reinforcements before concrete casting to replace concrete in the middle of the slab, which exerts little influence upon the flexural capacity (refer to Figure 1). Mike [2] argued that voided slabs

with segmented void shaper can reduce the slab weight by as much as 35% compared to a RC solid slab with the same flexural strength and maintain the flexural strength equivalent to a RC solid slab with the same thickness, theoretically. As reported by Ibrahim et al. [3], a voided slab with spherical voids behaved like a conventional two-way solid slab, and the voided slab carried 89–100% of the ultimate load of a solid slab with the same thickness. For the advantages of voided slabs using segmented void shaper, the concept and practice of voided slabs have been used, and several types of voided slabs have been developed currently.

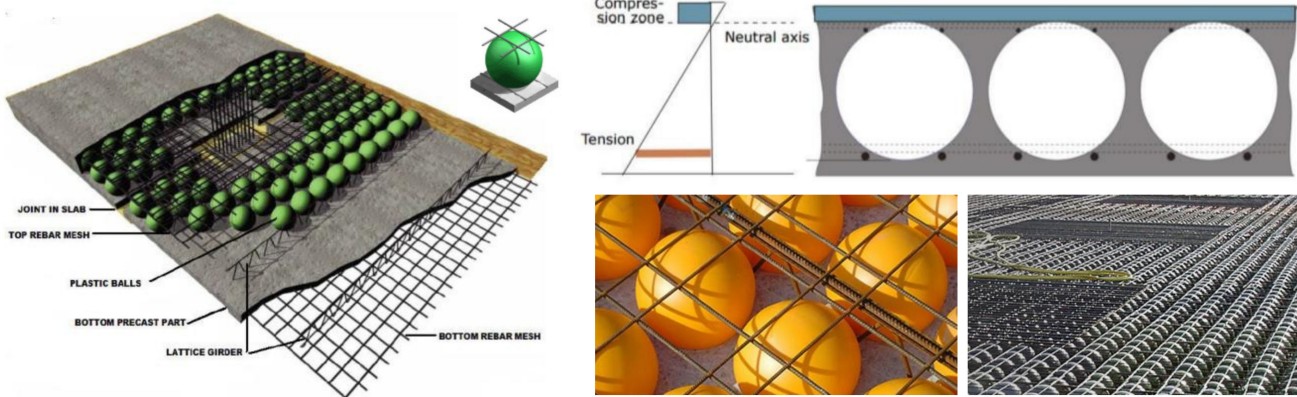

**Figure 1.** The concept of voided slab system and their application (BubbleDeck) [1].

Many tests have been conducted to evaluate the flexural capacities of voided slabs. According to the previous studies conducted by BubbleDeck [1], Kim et al. [4] and Lee et al. [5], the flexural capacities of the voided slabs, such as flexural strength, stiffness and ductility, were lower than that of the solid slabs with the same condition, even if some of specimen failed with shear (refer to Table 1). The studies pointed out that the shape and size of voids influenced the flexural capacities of voided slabs. Nimnin and Zain [6] demonstrated that the void shape significantly influenced the flexural strength of voided slabs, and spherical voids gave better results than the cubical ones. Corey [7] reported that the flexural strength of a voided slab with spherical voids is the same as that of a solid slab with equal depth if the compression zone used to apply the bending force to the slab sections does not enter the void zone, and the flexural stiffness of this voided slab is approximately 80–90% of that of the solid slab due to the cross-sectional loss caused by voids. Wondwosen [8] conducted finite element analysis of a voided slab with spherical voids and reported that the in-plane bending stiffness of the voided slab decreased by 20% compared to that of a solid slab with equal depth. Al-Gasham et al. [9] reported that the flexural stiffness and ductility of voided slabs with plastic ball decrease as the ratio of ball diameter to slab depth increase and argued that the reduction in the flexural stiffness of voided slabs could be attributed the direct reduction in the moment of inertia, fast enlargement of cracks and the decrease in bond strength of rebar due to voids. According to the results of these previous studies, the void shape is an important factor affecting the flexural capacity. However, the effect of the void shape on the flexural capacity of voided slab has not been clearly verified yet, and especially, the hole in the void shaper, in which focused on this study, has not been addressed as a variable.

## 1.2. Research Scope and Objectives

In this study, an analytical and experimental study were conducted to derive the optimal void shape and details focused on the flexural capacities of voided slabs. In particular, this study considered the hole of void shaper as an influence variable and focused on investigating the effect of that on the flexural capacity of the voided slab. Based on the finite element (FE) analysis, a donut-type void shaper, which was a hexahedron with rounded edges and a hole penetrating the center, was suggested as the optimal shape in

voided slabs, and experimental study was conducted to verify flexural capacities of the donut-type voided slab, focusing on the influence of void shape and fixing method of void shaper. The results of voided slabs were comprehensively compared to those of a solid slab specimen in five terms: failure behavior, flexural strength, deflection, stiffness and ductility. Based on the test results, a flexural design method of donut-type voided slab associated with void shape and fixing device of void shaper was suggested, and it was confirmed that the donut-type voided slab is one of the efficient alternatives to replace heavy flat plate slab.

**Table 1.** Flexural capacities of the current voided slab system.

| Researcher | Shape of Void | Ratio of Voided Slab to Solid Slab (%) | | Displacement Ductility Ratio ($\mu$) * |
|---|---|---|---|---|
| | | Flexural Strength | Flexural Stiffness | |
| BubbleDeck [1] | Spherical | 91 | 87 | 2.7 |
| S.M Kim [4] | Capsule | 87~92 | 91~107 | 2.07 |
| W.S Lee [5] | Spherical | 93.4 | 93 | 5.72 |
| | Oval | 72.3 | 81 | 4.17 |

* A measure of the ductility may be defined by the displacement ductility ratio.

## 2. Optimization of Void Shape

### 2.1. Parameters of Void Shape

To find out the parameters of void shape, the current void shapes were compared to each other. As a result, three parameters were found, such as base shapes, curvature radius of edge and hole diameter, as shown in Figure 2. According to previous studies [10–12], it was deduced that base shapes were related with the volume of the void. In case of the same height in the void, the volume of cuboid-shaped void was larger than that of sphere-shaped void. The volume of void was a key factor on voided slab, because it affected the self-weight of the slab directly. It was expected that the curvature radius of the edge influences a flexural strength and a ductility due to stress concentration at the edges, and because the curvature radius at the edge of the void became smaller, stress was concentrated more at the edge. It was also expected that the hole in voids was related with a flexural stiffness, because the center hole in the void shaper might improve the flexural stiffness by its good geometric shape, such as the reduction of the aspect ratio of the width and height of the void.

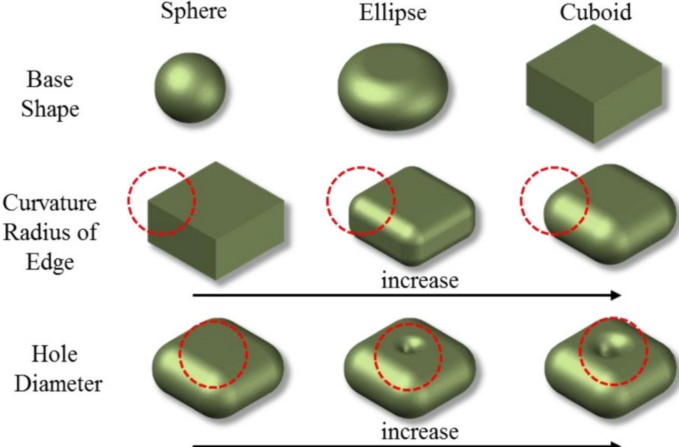

**Figure 2.** Parameters of the void shape.

Considering the parameters of the void shape, eight types of void shape were designed to find the optimal void shape by the nonlinear finite element analysis, as shown in Table 2.

The slab thickness was set to be 250 mm, having a heavy self-weight, and the width of the basic module of a voided slab was set to be 300 mm. The height of the void was assumed according to the depth of the compressive stress region with the maximum reinforcement ratio in the slab. The maximum depth of the compressive stress region was calculated by using the strain compatibility method. In order to prevent forming a void in the compressive stress region, the height of the void had to be less than 152 mm in a 250-mm-thick slab with 24 MPa of concrete and SD400 rebar. Taking into consideration the depth of the compressive stress region, slab thickness, workability such as rebar settlement and concrete casting, the height of the voids was set to be 140 mm, and the distance between them was 30 mm. Considering 300 mm in the basic module, the width or diameter of the voids was set to be 270 mm in all the voids except for the sphere-type void, whose width was decided by its height.

**Table 2.** Properties of the void shaper for the FE analysis model.

| Item | | Sphere | Mushroom | Ellipse | Rect. Donut (D = 50) | Rect. Donut (D = 30) | Round Rect (R = 70) | Round Rect (R = 50) | Square |
|---|---|---|---|---|---|---|---|---|---|
| Shape | |  |  |  |  |  |  |  |  |
| Base shape | | Sphere | Elliptical Sphere | | Cuboid | | | | |
| Radius curvature of edge | mm | 70 | 50 | 70 | 70 | 70 | 70 | 50 | 0 |
| Hole Diameter | mm | - | - | - | 50 | 30 | - | - | - |
| Volume | cm³ | 1436 | 5625 | 6300 | 7380 | 7650 | 7785 | 8910 | 10,125 |
| Width | mm | 140 | 270 | 270 | 270 | 270 | 270 | 270 | 270 |
| Height | mm | 140 | 140 | 140 | 140 | 140 | 140 | 140 | 140 |
| Weight reduction | % | 20.0 | 25.0 | 28.0 | 32.8 | 34.0 | 34.6 | 39.6 | 45.0 |

### 2.2. Finite Element Analysis Model of Voided Slab

Numerical simulations using nonlinear finite element analysis methods were conducted to derive the optimal void shape by using the FE analysis program named 'LUSAS' [13]. Generally, a 2D model was often used to conduct numerical simulations of the slabs by using finite element method when slabs were uniform in both longitudinal and transverse axes. However, in the case of voided slabs, it was impossible to use 2D model because of the irregular section geometry along the longitudinal and transverse axis due to voids. To consider complex void shapes precisely, a 3D model was used to generate concrete web parts between voids, and tetrahedral elements that had four nodes were used to generate FE meshes of the voided slabs that had extraordinary shapes of voids inside.

To conduct a nonlinear finite element analysis, two material models were used. The bilinear model, which assumed that the rebar behavior will be totally elastoplastic in the tensile and compressive loading conditions, was used for the rebar, and the LUSAS concrete model 94 [13], which can consider multi-cracks and strength softening of concrete, was used for concrete.

The properties of slab model were idealized as 8.9 m in length and 300 mm in width to conduct a FE analysis. It was enough to conduct a FE analysis with a line module model of slab to compare the effects of void shapes on flexural capacities, because voids are located with a uniform gap toward the width and length directions of the slab, and this way is more

time-efficient than using whole slab model. To simulate the real state of continuous slabs for building, the support conditions were set to be the fixed condition, and a distributed load was imposed. Loads, imposed on slabs, increased until the slab was destroyed in order of the self-weight of the slab, dead load and live load. Further information of the FE model is shown in Table 3.

**Table 3.** Properties of the FE analysis model.

| Rebar | Transverse direction | Top | D10 × 2 |
|---|---|---|---|
| | Longitudinal direction | Bottom | D13 × 2 |
| Mesh | Types of Mesh | Tetrahedral | |
| | Mesh size | 30 mm | |
| Size | Width | 300 mm | |
| | Height | 250 mm | |
| | Length | 8900 mm | |
| Load | Self-weight | 3.2~5.9 kN/m² | |
| | Dead load | 1.81 kN/m² | |
| | Live load | 1.96 kN/m² | |
| Material | $f_{ck}$ | 24 MPa | |
| | $f_y$ | 400 MPa | |
| Boundary condition | | Fixed end | |

The convergency test was conducted by mesh size to validate the FE model in terms of the accuracy of results and the time efficiency. The five tetrahedral elements with different mesh sizes of 10 mm, 20 mm, 30 mm, 50 mm and 80 mm were evaluated. The results converged into the theoretical ultimate load-bearing capacity of 32 kN/m² with less than a 5% difference when the mesh size was less than 30 mm, as shown in Figure 3. Therefore, a tetrahedral element with 30-mm mesh size was used in the FE analysis for voided slabs.

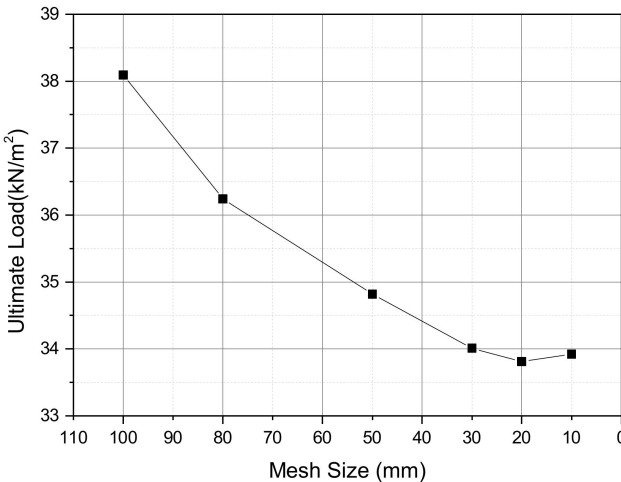

**Figure 3.** The results of the convergency test.

### 2.3. Finite Element Analysis Results and Discussion

To evaluate the structural capacity of voided slabs, the load-bearing capacity and the flexural stiffness were compared. A design load, an ultimate load and a residual load were compared in terms of a load-bearing capacity, and an initial stiffness, a failure stiffness and a transition stiffness were compared in terms of a flexural stiffness (refer to Figure 4). The design load is the sum of the dead load, live load and self-weight of the slab. The ultimate

load is a load at failure. The residual load is a difference between the design load and the ultimate load. The initial stiffness is a slope of a line through the origin and the point on the load–deflection curve at the design load. The failure stiffness is a slope of a line through the origin and the point on the load–deflection curve at the ultimate load. The transition stiffness is a slope of a line through the point on the load–deflection curve at the design load and the point on the load–deflection curve at the ultimate load.

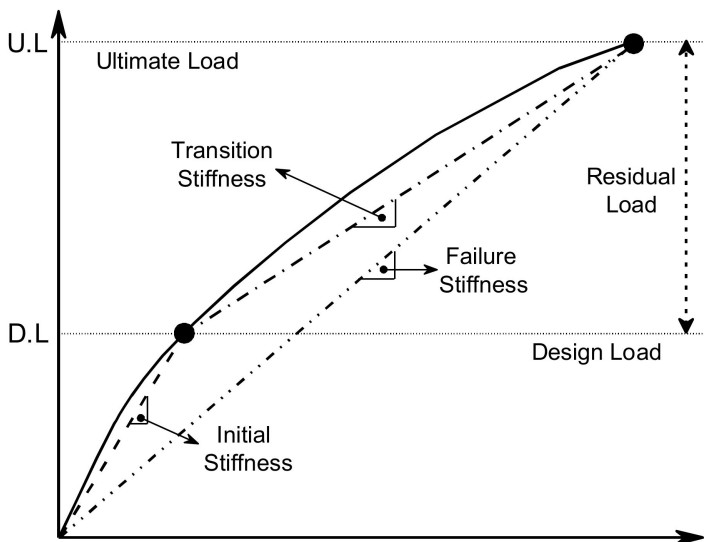

**Figure 4.** Definitions of Load and Stiffness.

Table 4 and Figure 5 show the results of the FE analysis of about eight cases of voided slabs and solid slab. All the slab models were found to be on the safety side under the design load and showed a general flexural behavior with some flexural cracks until the design load. The voided slabs except 'Square' and 'Round Rect (R = 50 mm)' showed flexural failure with the yield of bottom rebars at the ultimate load. On the other hand, the voided slabs with 'Square' and 'Round Rect (R = 50 mm)' failed with concrete crushing at the edge of void due to stress concentration at the ultimate load before the yield of rebars.

The design load was varied by the void shapes with its volume. The solid slab showed the largest design load of 9.70 kN/m². In the case of voided slab, the voided slab with 'Sphere' showed the largest design load of 9.29 kN/m², and the voided slab with 'Square' showed the smallest design load of 7.11 kN/m². Voided slabs showed the decrease in the design load by 4~27% compared to that of the solid slab. The ultimate load decreased in the cases of void shape such as 'Round Rect (R = 50 mm)' and 'Square'. The solid slab showed the ultimate load of 34.01 kN/m². The voided slabs, except 'Round Rect (R = 50 mm)' and 'Square', showed the similar ultimate load compared to the solid slab. The voided slabs with 'Round Rect (R = 50 mm)' and 'Square' showed the ultimate loads of 32.08 kN/m² and 28.50 kN/m², respectively. These were 94% and 84% of that of the sold slab, respectively. The residual load was varied by the void shapes in contrast with the ultimate load. The solid slab showed the residual load of 24.31 kN/m². In the case of voided slab, the voided slab with 'Rect Donuts (D = 50 mm)' showed the largest load-bearing capacity of 26.17 kN/m² and the voided slab with 'Square' showed the smallest load-bearing capacity of 21.39 kN/m². The voided slabs except in the cases of void shape such as 'Round Rect (R = 50 mm)' and 'Square' showed the increase in the residual load by up to 8% compared to that of the solid slab.

The deflection at the design load was varied by void shapes. The solid slab and the voided slab with 'Sphere' showed the largest deflection of 2.80 mm at the design load, and the voided slabs with 'Square' showed the smallest deflection of 2.34 mm at the design load. The voided slabs showed the decrease in the deflection at the design load by up to 16% compared to that of the solid slab. The deflection at the ultimate load was also

varied by the void shapes. The solid slab showed the smallest deflection of 22.42 mm at the ultimate load, except the voided slab with 'Square', which failed with concrete crushing. Among the voided slabs failed with the yield of bottom rebars, the voided slab with 'Round Rect (R = 70 mm)' showed the largest deflection of 26.50 mm at the ultimate load, and the voided slabs with 'Sphere' showed the smallest deflection of 22.32 mm at the ultimate load. The deflection of voided slabs at the ultimate load tended to increase in the deflection by up to 18% compared to that of the solid slab when the yield of rebar occurred.

The change of flexural stiffness according to void shape was similar to the change of deflection, because the flexural stiffness related with the deflection. However, the flexural stiffness was also evaluated, because the flexural stiffness also related with the load-bearing capacity. The solid slab showed the largest initial stiffness of 3.45. In the case of voided slab, the voided slab with 'Sphere' showed the largest initial stiffness of 3.30, and the voided slab with 'Square' showed the smallest initial stiffness of 3.02. The voided slabs showed the decrease in the initial stiffness by up to 12% compared to that of the solid slab. The failure stiffness was also varied by the void shapes. The solid slab showed the largest failure stiffness of 1.52. In the case of voided slab, the voided slab with 'Sphere' showed the largest initial stiffness of 1.45, and the voided slab with 'Round Rect (R = 50 mm)' showed the smallest initial stiffness of 1.26. The voided slabs showed the decrease in the failure stiffness by up to 17% compared to that of the solid slab. The transition stiffness of the voided slab was investigated for evaluating the change of flexural stiffness more precisely owing to the increase of the load after the design load. The transition stiffness was varied by the void shapes in contrast with the ultimate load. The solid slab showed the largest transition stiffness of 1.24. In the case of a voided slab, the voided slab with 'Rect Donuts (D = 50 mm)' showed the largest transition stiffness of 1.22, and the voided slab with 'Square' showed the smallest transition stiffness of 1.07. The voided slabs showed the decrease in the transition stiffness by up to 14% compared to that of the solid slab.

**Table 4.** The results of the FE analysis.

| Division | Solid | Sphere | Mushroom | Ellipse | Rect Donuts (D = 50) | Rect Donuts (D = 30) | Round Rect (R = 70) | Round Rect (R = 50) | Square |
|---|---|---|---|---|---|---|---|---|---|
| Self-weight (kN/m$^2$) | 5.89 | 5.48 | 4.41 | 4.24 | 3.96 | 3.88 | 3.85 | 3.56 | 3.3 |
| Design Load (kN/m$^2$) | 9.70 | 9.29 | 8.22 | 8.05 | 7.77 | 7.69 | 7.66 | 7.37 | 7.11 |
| Ultimate Load (kN/m$^2$) | 34.01 | 33.8 | 33.78 | 33.83 | 33.94 | 33.81 | 33.81 | 32.08 | 28.50 |
| Residual Load (kN/m$^2$) | 24.31 | 24.51 | 25.2 | 25.78 | 26.17 | 26.12 | 26.15 | 24.71 | 21.39 |
| Deflection at D.L (mm) | 2.80 | 2.80 | 2.52 | 2.58 | 2.49 | 2.47 | 2.46 | 2.41 | 2.34 |
| Deflection at U.L (mm) | 22.42 | 23.32 | 25.2 | 25.78 | 24.03 | 25.55 | 26.5 | 25.42 | 22.3 |
| Initial Stiffness | 3.45 | 3.30 | 3.25 | 3.10 | 3.10 | 3.10 | 3.10 | 3.04 | 3.02 |
| Failure Stiffness | 1.52 | 1.45 | 1.34 | 1.31 | 1.41 | 1.32 | 1.28 | 1.26 | 1.28 |
| Transition Stiffness | 1.24 | 1.20 | 1.13 | 1.11 | 1.22 | 1.13 | 1.09 | 1.08 | 1.07 |
| Failure mode at U.L | Y | Y | Y | Y | Y | Y | Y | C | C |

Note: 'Y' is the failure with the yield of rebars, and 'C' is the failure with the concrete crushing at the edge of void; 'U.L' is the load at the failure, and 'D.L' is the design load.

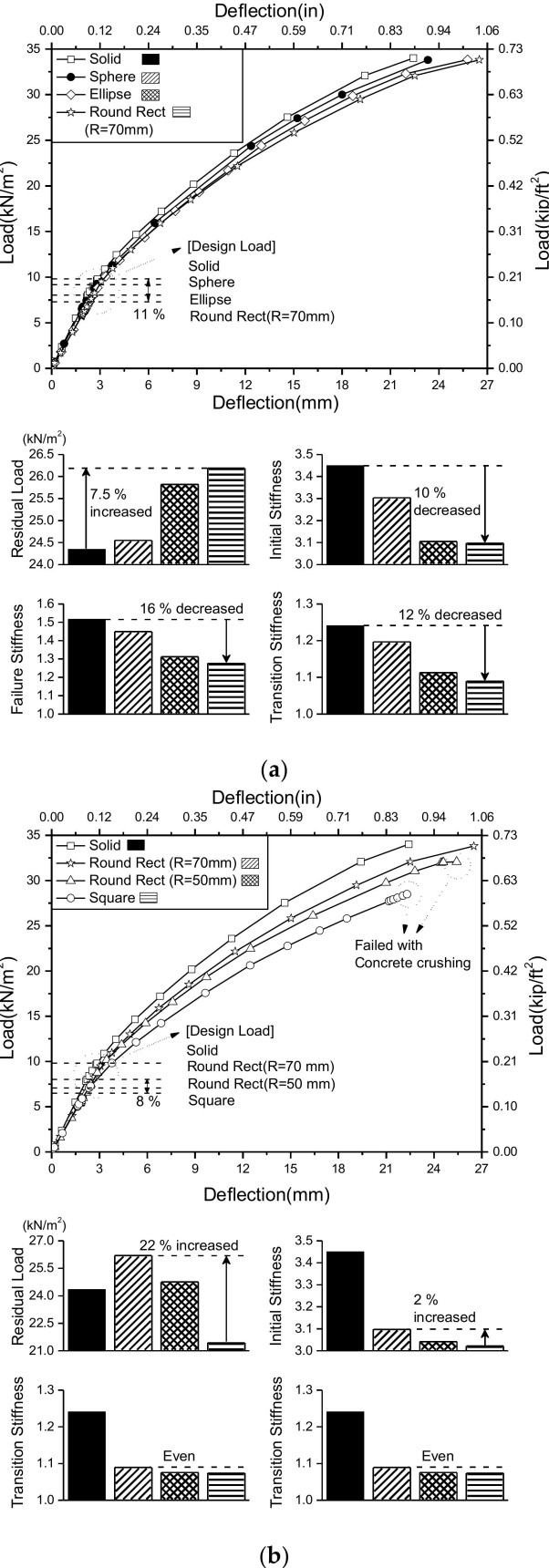

**Figure 5.** *Cont.*

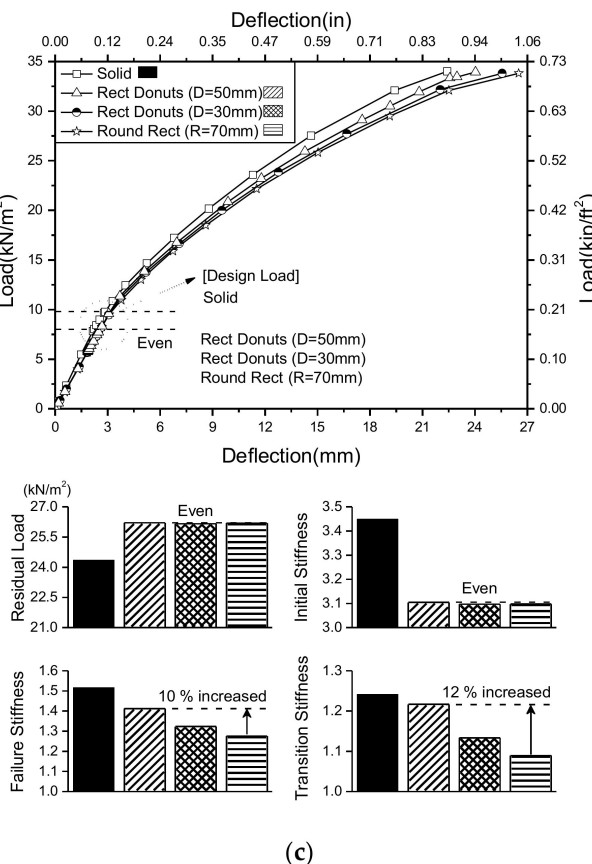

**Figure 5.** Load–deflection relationship of voided slab according to the parameters: (**a**) the base shape of void; (**b**) the radius curvature of edge; (**c**) the hole diameter.

Based on FE analysis results, the effect of each void shape parameter was investigated. The base shape of void was closely related with the volume of void, not the section properties of voided slab. Therefore, it was indicated that the base shape of void mainly influenced the load-bearing capacity of voided slab, as shown in Figure 5a. The curvature radius of edge was closely related with the stress concentration at the edge of void, not the section properties of the voided slab. The FE analysis also showed a concrete crushing failure at the edge of void before the yield of bottom rebars owing to stress concentration when the curvature radius of edge decreased less than 70 mm. Therefore, it was indicated that the curvature radius of edge of void influenced the load-bearing capacity and failure mode of the voided slab, as shown in Figure 5b. The hole diameter was related with the flexural stiffness of the voided slab. Among the flexural stiffness, the hole diameter mainly influenced the transition stiffness. Therefore, it was indicated that the hole diameter influences the flexural stiffness after the crack occurred, as shown in Figure 5c.

### 2.4. Determination of the Optimal Void Shape

Based on above FE analysis results, it was deduced that the self-weight reduction of a slab was mainly decided by the base shape of void. Thus, the choice of base shape is important to decide the optimal void shape, because the self-weight reduction is the major reason for the use of voided slab system in buildings. The direct evaluation of the base shapes was not conducted for the determination of the optimal void shape in this study, because the proper self-weight reduction would be varied case by case. However, the effect of the base shapes on the flexural strength should be considered for the determination of the optimal void shape, because the base shape also influences the flexural strength. The voided slab can secure the same or more level of the flexural strength by proper application of the base shape and the curvature radius of edge compared to solid slab. In addition, the

lack of flexural stiffness of voided slab due to the base shape and the curvature radius of edge can be complemented partly by the hole in void. Thus, when all these considerations take into account, it is expected that the optimal void shape has the base shape of cuboid, the curvature radius of edge by 70 mm and the hole in void shaper.

To verify the hypothesis, the above void shapes were compared to each other again focused on the load-bearing capacity and the flexural stiffness. The residual load-bearing capacity was compared in terms of the load-bearing capacity of voided slab, and the deflection at 20.4 kN/m$^2$, which was the load corresponded to $0.6M_y$ was compared in terms of the flexural stiffness. The reasons of comparing these values are as follows. It is not reasonable to use the ultimate load for comparing the load-bearing capacity of voided slab because the self-weight of voided slabs was varied according to void shapes, and the self-weight influenced the ultimate load-bearing capacity. As the difference between the design load and the ultimate load means the rest of load-bearing capacity of slab after the design load, the residual load should be used to compare the load-bearing capacity instead of the ultimate load. Therefore, the residual load-bearing capacities were used as the criterion to determine the optimal void shape. In addition, voided slabs were vulnerable to flexural stiffness due to the loss of cross-section area, and the flexural stiffness could be evaluated by the deflection. For comparing the flexural stiffness by the deflection, the deflections at a certain load should be compared. The limit of deflection is usually evaluated at a service load, and the service load is generally assumed as $0.6M_y$. Therefore, the deflections at 20.4 kN/m$^2$ were also used as the criterion to decide the optimal void shape.

The sequential elimination method was used to decide the optimal voids shape with these two criterions. The sequential elimination method is a one of reasonable method to select the one from among several. The concept of this method is the eliminating something sequentially according to a certain criterion. The eliminating procedure is performed repeatedly until only the one thing is left in the order of importance of the criterions. Thus, the order of importance of criterions should be defined. In this study, it was assumed that the load-bearing capacity of a voided slab was more important than the flexural stiffness, because the load-bearing capacity closely related with the safety of the voided slab system. Following these rules, the top three void shapes were selected among the above eight types of void shape, and the others were eliminated according to the residual load-bearing capacity in terms of the load-bearing capacity. After that, a certain void shape, which showed the smallest deflection, was decided as the optimal void shape from among the rest of three void shapes in the former eliminating process.

Figure 6a showed the results of the first eliminating process of the sequential elimination method. The voided slab with 'Rect Donuts (D = 50)', 'Rect Donuts (D = 30)' and 'Round Rect (R = 70)' showed the largest values of the residual load among the above eight types of void shape. Therefore, these void shapes were selected as the top three in terms of the load-bearing capacity, and the other void shapes were eliminated through the first sequential elimination process. Figure 6b showed the results of the second eliminating process in terms of the flexural stiffness. Among the three remaining void shapes through the first eliminating process, the voided slab with 'Rect Donuts (D = 50)' had the smallest deflection of 9.5 mm at the service load. As a result, 'Rect Donuts (D = 50)' was decided as the optimal void shape in this study. It showed good load-bearing capacity with the residual load-bearing capacity of 26.12 kN/m$^2$ and the ultimate load-bearing capacity of 33.94 kN/m$^2$, which were 108% and 100% of their counterpart with the solid slab, respectively. In addition, it also showed good flexural stiffness at the service load, which was 94% of their counterpart with the solid slab, and a good self-weight reduction ratio more than 30%.

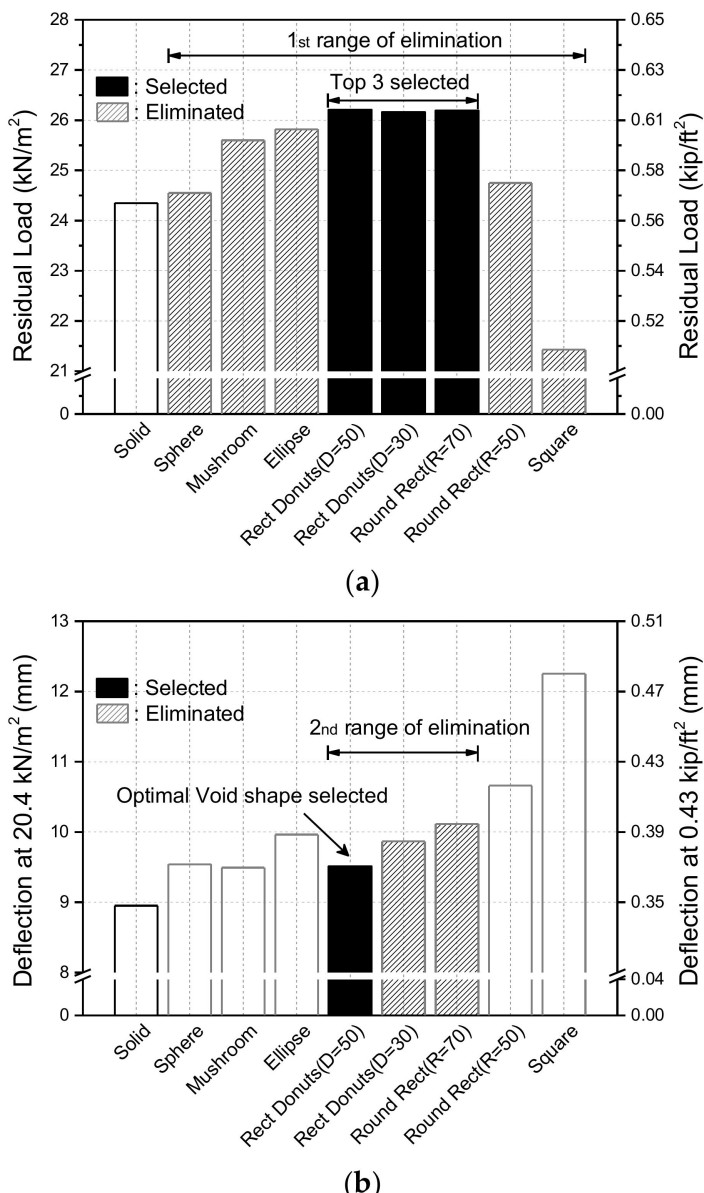

**Figure 6.** Determination of the optimal void shape. (**a**) by the load-bearing capacity (1st try); (**b**) by the load-bearing capacity (2nd try).

## 3. Experimental Program

### 3.1. Configuration of Donut-Type Voided Slab Specimens

The objective of the flexural test was to evaluate the flexural capacity of donut-type voided slab by comparing the failure behavior, ductility, flexural strength and stiffness with those of a solid slab and non-donut-type voided slab. The specific objective of the test was to verify the effect of the center hole and fixing methods of the void shapers.

To generate voids in the voided slab specimens, two types of void shapers were used: the donut-type void shaper and the non-donut-type void shaper, as shown in Figure 7. The donut-type void shaper was a hexahedron with rounded edges and a hole penetrating the center. Reflecting the results of the FE model, the void height and width were 140 mm and 270 mm, respectively. The hole diameter was 50 mm, and the distance between the voids was set to 30 mm in both the longitudinal and transverse directions. The non-donut-type void shaper has the same specifications as the donut-type void shaper, but there is no hole penetrating the center.

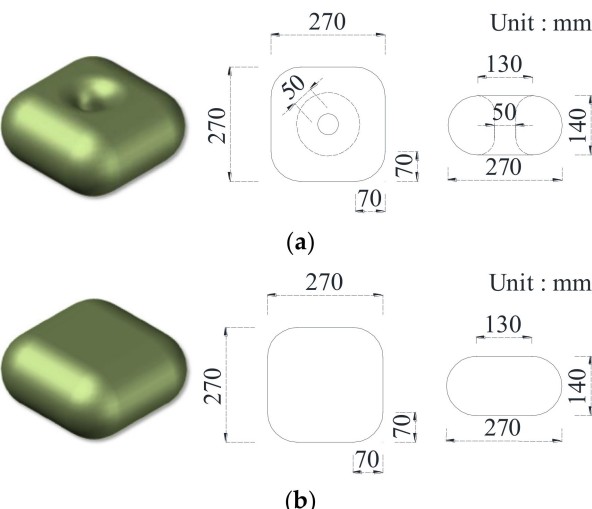

**Figure 7.** Details of donut and non-donut-type void shaper: (**a**) donut-type void shaper; (**b**) non-donut-type void shaper.

To hold the donut-type void shapers in place, keeping them in the center of the slab's depth, two types of fixing methods were used: the spacer and the merged type, as shown in Figure 8. The spacer type consisted of void shaper with protrusions, which functioned as the spacers between the top and bottom rebars without requiring additional steel cages to hold the void shapers. The merged type fixing method holds void shapers using a steel cage, which was fabricated by welding the top and bottom rebars with D6 diagonal rebars.

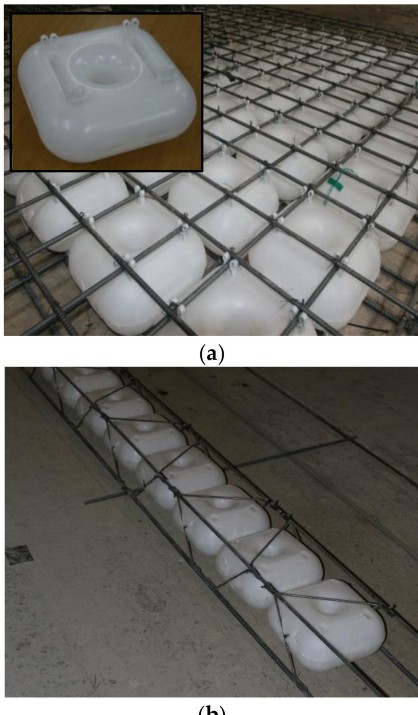

**Figure 8.** Fixing methods of the donut-type voided slab: (**a**) spacer-type fixing method; (**b**) merged-type fixing method.

Four slab specimens were designed to investigate the effect of void shape and the fixing method: a conventional solid reinforced concrete slab (Solid), a donut-type voided slabs with the spacer-type fixing method (OF-V-S-D), a donut-type voided slabs with the merged-type fixing method (OF-V-M-D) and a non-donut-type voided slabs with the

spacer-type fixing method (OF-V-S-R). The widths and lengths of the slab specimens were 1250 mm and 3300 mm, respectively. The thicknesses of slab were 250 mm. In general, voided slabs are vulnerable to shear strength deterioration; hence, the slab specimens were designed to have low tensile reinforcement ratios (ρ) of 0.384% to induce flexural failure prior to shear failure. The merged-type fixing device was placed in the longitudinal direction of the specimen. Detailed specifications of the specimens are presented in Table 5 and Figure 9.

**Table 5.** Details of the specimens.

| Name | Height (mm) | Width (mm) | Length (mm) | Clear Span (mm) | Top Rebar | Bottom Rebar | Tensile Reinforcement Ratio (%) | Void Shape | Fixing Type |
|---|---|---|---|---|---|---|---|---|---|
| Solid | 250 | 1250 | 3300 | 2850 | 8-D10 | 8-D13 | 0.384 | - | - |
| OF-V-S-D | 250 | 1250 | 3300 | 2850 | 8-D10 | 8-D13 | 0.384 | Donut | Spacer |
| OF-V-M-D | 250 | 1250 | 3300 | 2850 | 8-D10 | 8-D13 | 0.384 | Donut | Merged |
| OF-V-S-R | 250 | 1250 | 3300 | 2850 | 8-D10 | 8-D13 | 0.384 | Non-Donut | Spacer |

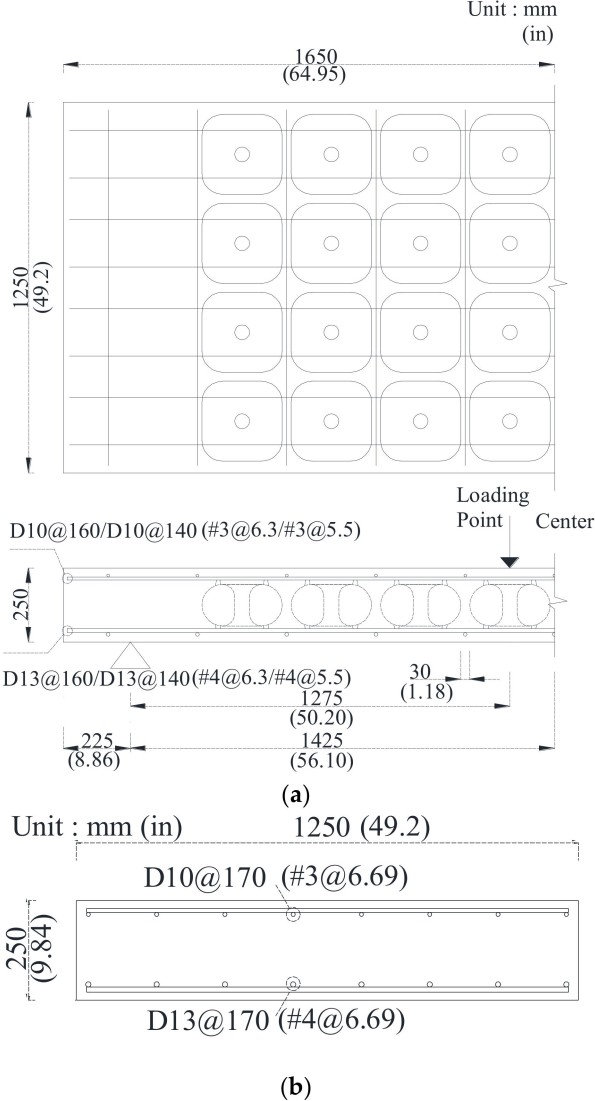

**Figure 9.** *Cont.*

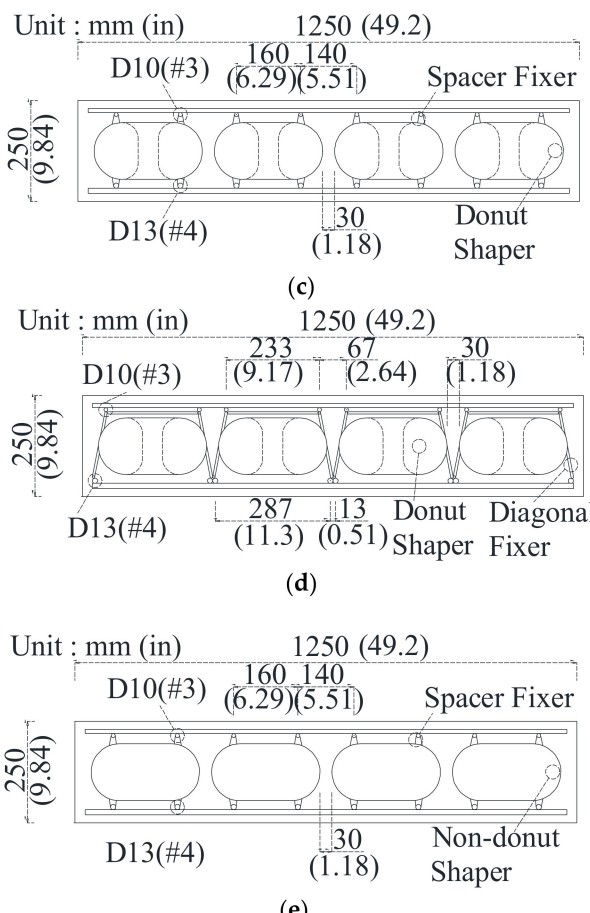

**Figure 9.** Details of specimens: (**a**) plan and elevation; (**b**) solid; (**c**) OF-V-S-D; (**d**) OF-V-M-D; (**e**) OF-V-S-R.

### 3.2. Materials of Specimens

The concrete used in all the slab specimens came from one batch. The design strength of the concrete was 24 MPa, and the mixing ratio is summarized in Table 6. Nine concrete cylindrical specimens were made with dimensions of 100 mm (diameter) × 200 mm (height) and then cured under the same conditions as that of the slab specimens. The concrete strength test conducted immediately before the structural test showed an average strength of 25.6 MPa, which was slightly higher than the design strength of 24 MPa.

**Table 6.** Mix proportion of concrete and material test results of rebars.

| Design Strength, (MPa) | W/C (%) | S/a (%) | Weight Ratio (kg/m³) | | | | |
|---|---|---|---|---|---|---|---|
| | | | Water | Cement | Sand | Coarse Aggregate | Admixture |
| 24 | 56.6 | 47.4 | 193 | 341 | 837 | 985 | 1.7 |
| Rebar Type | Nominal Strength, (Mpa) | | Yield Strength, (Mpa) | Tensile Strength, (Mpa) | | Elongation (%) | Elastic Modulus, (Gpa) |
| D6 | 440 | | 528 | 675 | | 15.28 | 205.8 |
| D10 | 400 | | 469 | 648 | | 17.36 | 196.1 |
| D13 | 400 | | 473 | 665 | | 18.21 | 194.3 |

For the rebar, D10 and D13 rebar with yield strength grades of 400 Mpa were used as the top and bottom rebars, and D6 indented rebar with a yield strength grade of 440 Mpa was used to fabricate the merged-type fixing devices. Tensile tests were conducted on the rebars, and the results are summarized in Table 6.

### 3.3. Loading and Measurement Set-Up

The tests were conducted by four-point bending test with two simple supports and two loading points, as shown in Figure 10. The clear span of specimens was 2850 mm, and the loading hinges were located 150 mm apart from the center to provide shear–span ratio of 6.0 to induce flexural failure. Loading was implemented with a 2000-kN static-dynamic hydro-actuator with a loading speed of 1 mm/min. The deflection was measured by seven linear variable differential transformers (LVDT) placed under the center, loading points and 1/4 points of the specimen, as shown in Figure 10. The strain gauges were placed at the center and loading points of the tension rebars in which the maximum moment would occur to measure the yield strength of the specimens.

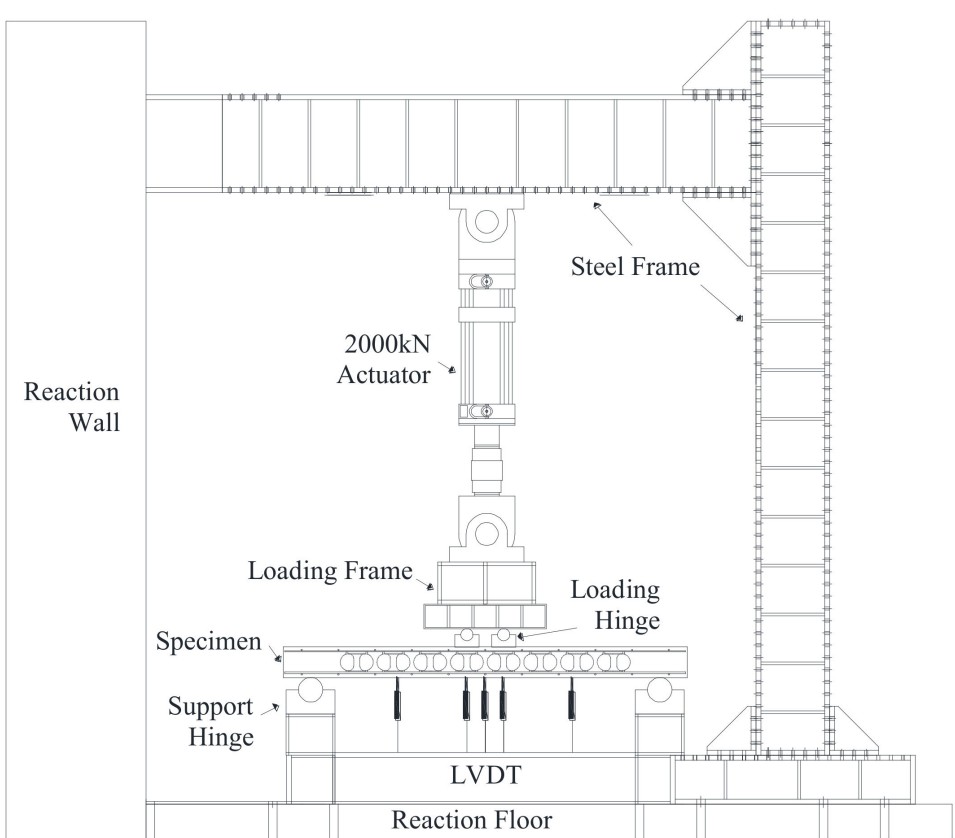

**Figure 10.** Test set-up.

## 4. Test Results and Discussion

### 4.1. Failure Behaviour and Crack Pattern

Figure 11 shows the load–deflection curve of specimen, and Figure 12 shows the crack pattern of specimen after the failure. The donut-type voided slab specimens showed a ductile flexural failure mode such as 'Solid', regardless of the fixing method. However, the non-donut-type voided slab specimen (OF-V-S-R) failed with the inclined flexural–shear crack opening after the yield of bottom rebar without fully crushing of concrete in compression zone. It was caused by 'OF-V-S-R' having a lower shear strength than flexural strength.

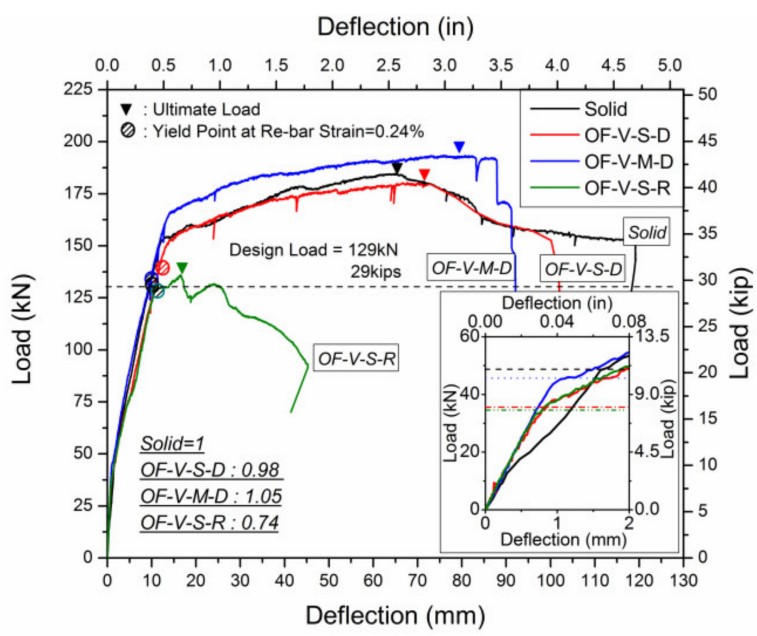

**Figure 11.** Load–deflection relationship of the test specimens.

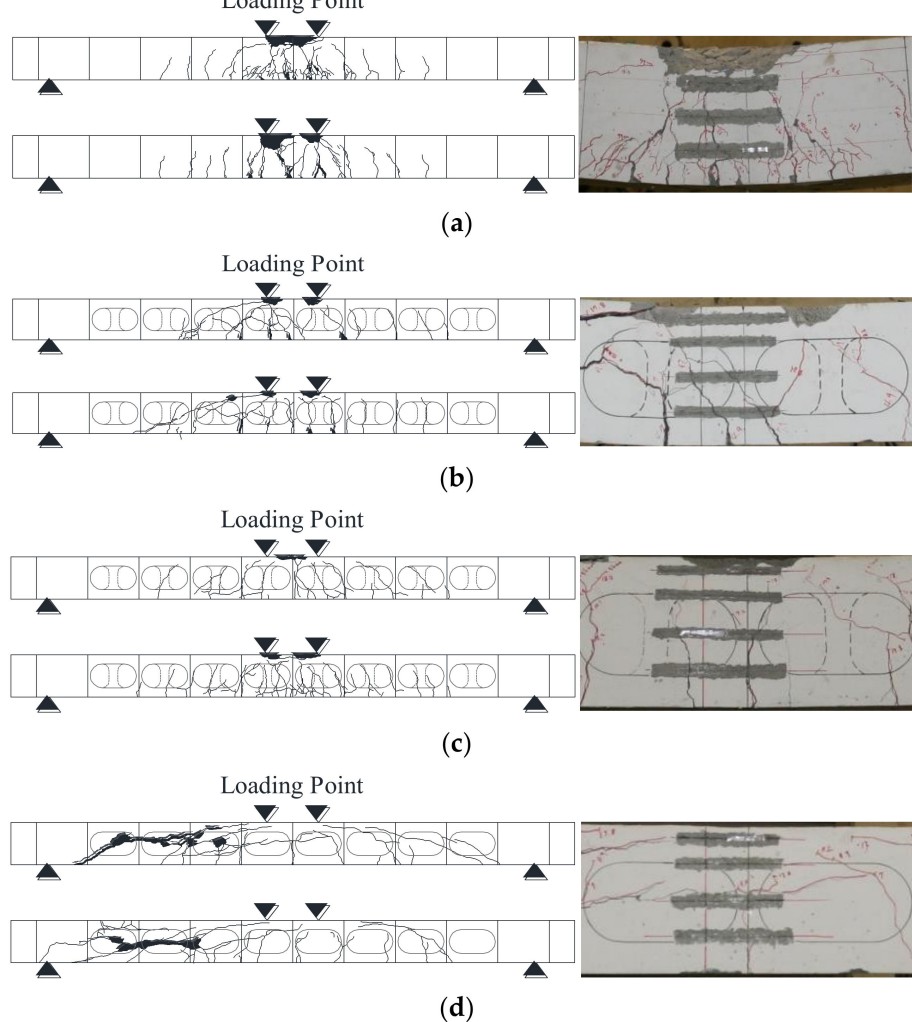

**Figure 12.** Crack pattern and failure mode. (**a**) Solid; (**b**) OF-V-S-D; (**c**) OF-V-M-D; (**d**) OF-V-S-R.

In 'Solid', flexural cracks were first observed in the center of lower part of specimen at a load of 48.41 kN. With the increase of the load, flexural cracks spread into upper parts and diffused from the center to the supports gradually. In addition to major cracks at the center, multiple hair cracks were also observed. The specimen failed with the crushing of upper concrete at the ultimate load after the yield of bottom rebar.

In 'OF-V-M-D', a donut-type voided slab with the merged-type fixing method, flexural cracks were firstly observed in the center of lower part of specimen at a load of 45.50 kN, and the failure behavior and crack pattern were similar to 'Solid'. With the increase of the load, flexural cracks spread into the upper parts and diffused from the center to the supports gradually. In addition to major cracks at the center, multiple hair cracks were observed. The specimen also failed with the crushing of upper concrete at the ultimate load after the yield of bottom rebar.

In 'OF-V-S-D', donut-type voided slab with the spacer-type fixing method, flexural cracks were firstly observed at relatively lower load of 36.69 kN, the failure behavior and crack pattern were different to 'Solid' and 'OF-V-M-D'. As the increase of the load, major cracks spread into upper parts, and diffused from the center to the supports gradually, while significantly fewer hair cracks occurred. In addition, flexural–shear cracks in a diagonal direction occurred at the end of test after the yield of bottom rebar and ultimate flexural strength, and the region of concrete crushing was smaller than that of 'Solid' and 'OF-V-M-D'. It seemed to be caused by the reduction of shear strength by voids.

In 'OF-V-S-R', non-donut-type voided slab with the spacer-type fixing method, flexural cracks were firstly observed at a load of 36.29 kN, which was similar to that of 'OF-V-S-D'. With the increase of the load, major flexural cracks gradually propagated with few hair cracks until bottom rebar yielding. After the yield of rebar, flexural-shear cracks suddenly occurred, and the specimen failed in shear. It was deduced that 'OF-V-S-R' had a lower shear strength than flexural strength.

### 4.2. Flexural Cracking Load of Donut-Type Voided Slab

Flexural cracking strength, related with a flexural cracking load directly, is important to define minimum tensile reinforcement ratio for slabs. Flexural cracking strength can be obtained from Equation (1) in ACI 318-11 [14], and flexural cracking strength should be higher in 'Solid' than the voided slab specimens theoretically, because these have lower values of gross concrete section moment of inertia ($I_g$) due to voids.

$$M_{cr} = \frac{f_r I_g}{y_t} \tag{1}$$

- $M_{cr}$ : Cracking moment (MPa)
- $I_g$: Moment of inertia of gross concrete section about centroidal axis
- $f_r$: Modulus of rupture of concrete
- $y_t$: Distance from centroidal axis of gross section to extreme tension fiber, neglecting reinforcement

However, because the donut-type voided slab has two types of cross-sections which can cause initial flexural crack (the minimum cross-section and the donut-type cross-section, as shown in Figure 13), the cross-section where the initial flexural crack occurs must be defined in order to use Equation (1). Therefore, to define the cross-section for flexural cracking strength of the donut-type voided slab, the test results were compared to the values calculated by Equation (1) in the two types of cross-sections (refer to Table 7). As expected, the initial flexural crack was occurred at a lower load in 'OF-V-S-D' and 'OF-V-S-R' than 'Solid', but contrary to expectations, the flexural cracking load of 'OF-V-M-D' was similar to 'Solid'. In addition, flexural cracking load of voided slabs derived from Equation (1) was higher than that of obtained from the test, and the difference was even greater in the 'Solid' specimen. It is deduced that the gap between these experimental values and calculated values resulted from the fact that the self-weight of specimen is not taken into

consideration. Practically, the uniformly distributed load of 7.36 kN/m, 5.41 kN/m and 5.38 kN/m already existed even before loading in the solid, donut and non-donut-type voided slab specimens, respectively, because of the self-weight of specimens. When the self-weight is taken into consideration, the flexural cracking load measured by the test is converted 60.13 kN for 'Solid', 54.11 kN for 'OF-V-M-D', 45.30 kN for 'OF-V-S-D' and 44.73 kN for 'OF-V-S-R'. As a result, it is clear that the flexural cracking load in donut and non-donut-type voided slabs is lower than that in a solid slab.

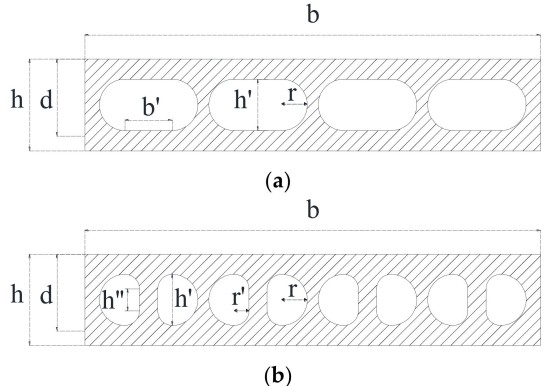

**Figure 13.** Types of cross-sections in donut-type voided slabs: (**a**) minimum cross-section; (**b**) donut cross-section.

**Table 7.** Test results.

| Specimen | Calculated Results | | | | Test Results | | | | | | Comparison | | |
|---|---|---|---|---|---|---|---|---|---|---|---|---|---|
| | $P_{cr,cal}$ * (kN) | $\delta_{cr,cal}$ † (mm) | $P_{n,cal}$ ‡ (kN) | $\delta_{n,cal}$ § (mm) | $P_{cr}$ £ (kN) | $\delta_{cr}$ € (mm) | $P_y$ ¥ (kN) | $\delta_y$ + (mm) | $P_{max}$ # (kN) | $\delta_{max}$ $ (mm) | $\frac{P_y}{P_n}$ | $\frac{P_{max}}{P_n}$ | $\frac{P_{max}}{P_{max,solid}}$ |
| Solid | 63.04 | 0.66 | 129.1 | 5.48 | 48.41 60.13 ᵂ | 1.6 | 134.01 145.73 ᵂ | 9.75 | 184.94 196.66 ᵂ | 64.77 | 1.04 | 1.43 | 1.0 |
| OF-V-S-D | 58.09 [55.51] | 0.66 [0.66] | 129.1 | 6.21 [6.58] | 36.69 45.30 ᵂ | 1.01 | 139.81 148.42 ᵂ | 11.76 | 180.25 188.86 ᵂ | 66.68 | 1.08 | 1.40 | 0.98 |
| OF-V-M-D (ρ = 0.036%) | 58.09 [55.51] | 0.66 [0.66] | 129.1 | 6.21 [6.58] | 45.50 54.11 ᵂ | 1.1 | 138.56 147.17 ᵂ | 10.70 | 193.43 202.04 ᵂ | 81.67 | 1.07 | 1.50 | 1.05 |
| OF-V-M-D (ρ = 0.043%) | | | 143.5 | 7.16 [7.51] | | | 143.62 152.23 ᵂ | 11.41 | | | 1.00 | 1.35 | |
| OF-V-S-R | 55.51 | 0.66 | 129.1 | 6.58 | 36.29 44.73 ᵂ | 0.99 | 129.04 137.48 ᵂ | 11.37 | 135.95 144.39 ᵂ | 16.54 | 1.00 | 1.05 | 0.74 |

[ ]: Values are calculated with the moment of inertial at the minimum cross-section area of the slab; ᵂ Values are considered with the self-weight of the specimen; * Cracking load, which is calculated with ACI 318-11; † Deflection at the cracking load, which is calculated with ACI 318-11; ‡ Nominal flexural load, which is calculated with ACI 318-11; § Deflection at the nominal flexural load, which is calculated with ACI 318-11; £ Measured load when the first crack occurred; € Measured deflection when the first crack occurred; ¥ Measured load when the bottom rebar yielded; + Measured deflection when the bottom rebar yielded; # Maximum load; $ Measured deflection at the maximum load.

Based on these results, it could be deduced that the hole in donut-type void shaper did not influence on flexural cracking strength significantly, but the fixing method of void shaper influence on the flexural cracking strength. In addition, it was verified that minimum cross-section properties should be used to calculate the flexural cracking moment in donut-type voided slabs, which are 81.6%~97.4% of what is obtained from Equation (1).

*4.3. Yield Flexural Strength of Donut-Type Voided Slab*

The yield point of specimens was defined by three methods. The one is when the two strain gauges placed at the center of bottom rebar reach 0.24%, which is the yield strain of rebar derived from material test. The yield load based on the strain of bottom rebar was 134.0 kN, 139.8 kN, 138.6 kN and 129.0 kN in 'Solid', 'OF-V-S-D', 'OF-V-M-D'

and 'OF-V-S-R', respectively. The difference of 4.6~5.8 kN in the yield load between solid and donut-type voided slab specimens seems to be caused by the difference in self-weight of slab, as above mentioned. Converting the moment at the center of slabs caused by the difference in self-weight into the load at the loading points provides the difference of 3.1 kN, which is close to the difference in the measured yield load. However, the method of defining yield point based on the strain of bottom rebar did not accord with the points of rapid deterioration in stiffness. Thus, the 'Park method' and 'Offset method' were additionally used to find the yield point [15]. Table 8 and Figure 14 show the yield point obtained by these methods. It is shown that both of methods provided slightly higher yield loads than that derived from the strain of bottom rebar. In particular, the 'Park method' find the points of rapid deterioration in stiffness more clearly (refer to Figure 14). Based on the 'Park method', 'OF-V-M-D' demonstrated a superior yield strength to 'OF-V-S-D', which was inferred to be enabled by the legs of fixing steel cage made of $f_y$ (=440 MPa), because it remained elastic at the yield of the bottom rebar. Based on test results, it is deduced that donut-type voided slab secures same or more the yield strength of the solid slab.

### 4.4. Ultimate Flexural Strength of Donut-Type Voided Slab

The maximum load was investigated to evaluate ultimate flexural strength of donut-type voided slab. All specimens sufficiently satisfied the nominal flexural load of 129.1 kN, as shown in Figure 12 and Table 6. Nominal flexural strength ($M_n$) could be calculated by Equation (2) based on the strain compatibility method [14].

'OF-V-S-D' and 'OF-V-M-D' showed an ultimate load of 180.3 kN and 193.4 kN, equivalent to 98% and 105% of that of 'Solid', respectively. 'OF-V-S-R' showed an ultimate load of 135.9 kN, which is much lower than 'Solid'. However, it is meaningless to compare this value, because 'OF-V-S-R' failed with a shear crack before reaching the ultimate flexural load. Based on the test results, it is found that donut-type voided slabs secure the same ultimate flexural strength of solid slabs. In particular, 'OF-V-M-D' showed superior flexural strength to 'Solid' and 'OF-V-S-D', which is inferred to be enabled by the fixing steel cage of the merged-type fixing method. The four add legs of the fixing steel cage were welded to the bottom rebars to fix the void shapers, which had a sectional of 113.2 mm$^2$ and the reinforcement ratio of 0.007%. It enhances the flexural strength by 9.2 kN·m, and this value is equivalent to 14.4 kN when converted into the flexural load. It is similar to 13.1 kN, which is the difference in the ultimate strength between the two donut-type voided slab specimens with and without fixing the steel cage. This effect of the fixing steel cage is also shown in Figure 14. 'Solid' and 'OF-V-S-D' tolerated 10~15 kN after the yield of the bottom rebar, showing gradual deterioration in stiffness until it deteriorated rapidly at 150 kN. 'OF-V-M-D' tolerated approximately 26 kN after the yield of the bottom rebar, followed by the rapid deterioration in stiffness. It is caused by the legs of the fixing steel cage remaining elastic at the yield of the bottom rebar, as mentioned above. Therefore, it is deduced that the fixing steel cage improves the flexural strength of 'OF-V-M-D'. These results confirm that Equation (2) for the flexural strength of reinforced concrete slabs prescribed in the current design code [14] can be applied to the donut-type voided slabs with the spacer-type fixing method, whereas Equation (3) is appropriate to the calculation of flexure strength of the donut-type voided slabs with the merged-type fixing method, because it takes into consideration the legs of the fixing steel cage.

**Table 8.** Yield strength and displacement–ductility ratio.

| Specimen | Considering Strain of Bottom Rebar * | | Park Method † | | Offset Method ‡ | | Displacement Ductility ($\mu$) § | | |
| --- | --- | --- | --- | --- | --- | --- | --- | --- | --- |
| | $P_y$ (kN) | $\delta_y$ (mm) | $P_y$ (kN) | $\delta_y$ (mm) | $P_y$ (kN) | $\delta_y$ (mm) | Strain of Bottom Rebar | Park Method | Offset Method |
| Solid | 134.01 | 9.75 | 153.46 | 12.71 | 152.90 | 12.54 | 6.64 | 5.10 | 5.17 |
| OF-V-S-D | 139.81 | 11.76 | 154.10 | 15.56 | 150.95 | 14.01 | 5.67 | 4.29 | 4.76 |
| OF-V-M-D | 138.56 | 10.70 | 165.99 | 14.58 | 153.69 | 12.07 | 7.63 | 5.60 | 6.77 |
| OF-V-S-R | 129.04 | 11.37 | 127.00 | 10.56 | 119.21 | 9.51 | 1.45 | 1.57 | 1.74 |

* Values of $P_y$ and $\delta_y$ when strain gauges placed at the tensile bars at the center of the slab reach 0.24%; † The first straight line corresponds to the initial stiffness (K), defined between 0% and 75% of the peak load. The second line defined as horizontal line at peak load. The yield point on the load-deformation curve defined the projected line from the intersection point of these two lines. ‡ A line is constructed parallel to the initial portion of the stress–strain curve but offset by 0.002 in/in (0.2%) from the origin. The 0.2% offset yield strength is the stress at which the constructed line intersects the stress–strain; § $\mu = \delta_y / \delta_{max}$.

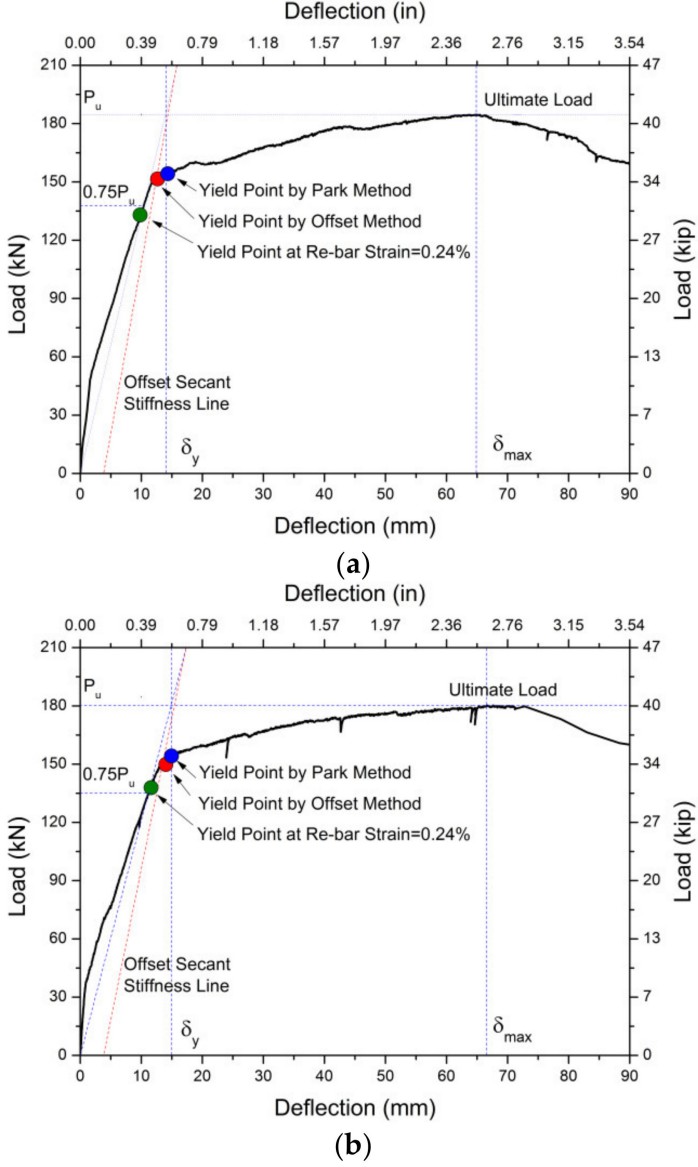

**Figure 14.** *Cont.*

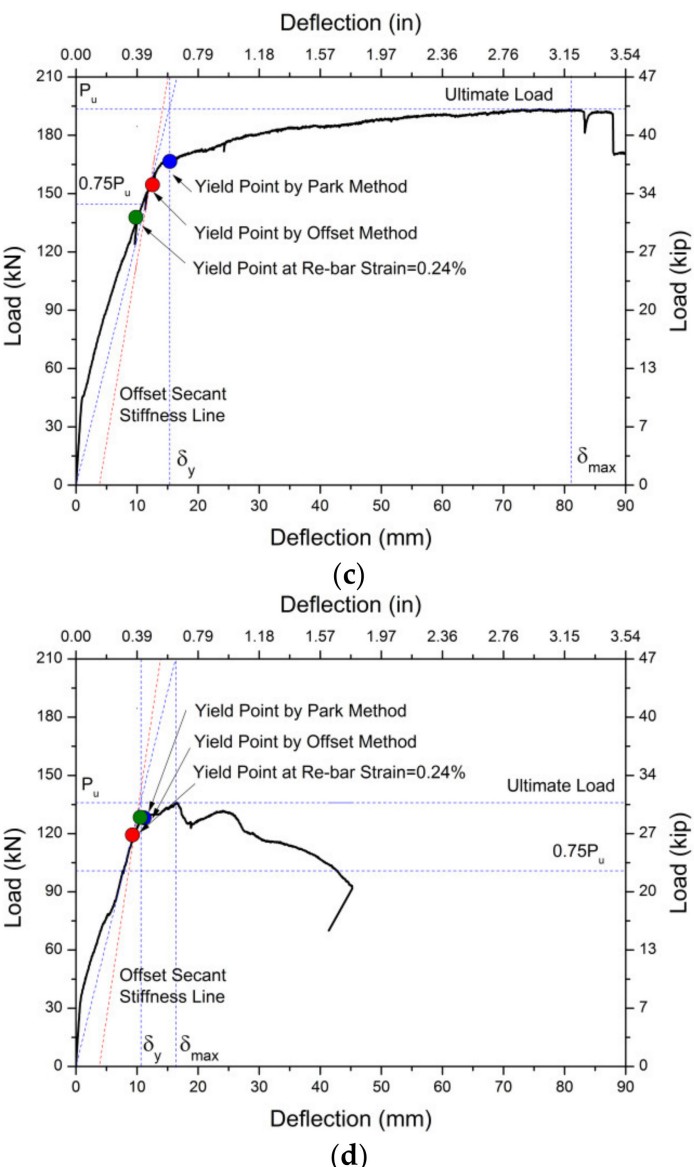

**Figure 14.** Yield points of the test specimens: (**a**) Solid; (**b**) OF-V-S-D; (**c**) OF-V-M-D; (**d**) OF-V-S-R.

$$M_n = A_s f_y \left( d - \frac{a}{2} \right) \tag{2}$$

$$M_n = (A_s + A_{legs}) f_y \left( d - \frac{a}{2} \right) \tag{3}$$

- $M_n$ : nominal flexural strength
- $A_s$: area of non-prestressed tension reinforcement
- $A_{legs}$: area of the legs of fixing steel cage
- $f_y$: specified yield strength of non-prestressed reinforcement
- $d$: effective depth of slab specimen
- $a$: depth of equivalent rectangular stress block

### 4.5. Ductility of Donut-Type Voided Slab

The ductility of the slab can be evaluated by the displacement ductility ratio ($\mu$). Displacement ductility, the criterion for the inelastic deformation of members, was defined as the ratio of the deflection at ultimate load to that at yield load, as shown in Equation (4).

$$\mu = \frac{\delta_u}{\delta_y} \tag{4}$$

- $\delta_u$: deflection at the ultimate load
- $\delta_y$: deflection at the yield load

The ductility of the donut-type voided slabs was similar or superior to that of the solid slab, as shown in Table 8. The displacement–ductility ratio by the 'Park method' was 5.10, 4.28, 5.60 and 1.56 in 'Solid', 'OF-V-S-D', 'OF-V-M-D' and 'OF-V-S-R', respectively. In the case of using other methods, as mentioned above, the donut-type voided specimens also showed a displacement–ductility ratio of more than four in order to perform as a general flexural member.

### 4.6. Flexural Stiffness of Donut-Type Voided Slab

In order to evaluate the flexural stiffness of the donut-type voided slabs, the flexural stiffness and moment of inertia were compared. For flexural stiffness, the secant stiffness at the cracking load and the stiffness between the cracking and yield points are compared. The theoretical moment of inertia of the gross concrete section, cracked section and effective moment of inertia derived from Equations (5)–(9) were compared with the experimental moment of inertia derived from Equation (10). The moment of inertia was calculated for the minimum section and donut section and compared with the test results. The moment of inertia at the minimum and donut sections can be easily calculated by subtracting $I_{gM}$ and $I_{gD}$ from $I_g$ of 'Solid', respectively (refer to Figure 13).

$$I_g = \frac{bh^3}{12} - N(I_{gM} \text{ or } I_{gD}) \tag{5}$$

$$I_{gM} = \left(\frac{b'h'^3}{12} + \frac{\pi r^4}{4}\right) \tag{6}$$

$$I_{gD} = \frac{\pi r^4}{4} + \frac{r'h''^3}{6} + \left(\frac{\pi r'^4}{4} + \pi r'^2 \left(\frac{h''}{2} + \frac{4r'}{3\pi}\right)^2\right) \tag{7}$$

$$I_{cr} = \frac{b(kd^3)}{3} + nA_s(d - kd)^2 \tag{8}$$

$$I_e = \left(\frac{M_{cr}}{M_a}\right)^3 I_g + \left[1 - \left(\frac{M_{cr}}{M_a}\right)^3\right] I_{cr} \tag{9}$$

$$I_{test} = \frac{Pl_a}{48E_c\delta}(3l^2 - 4l_a^2) \tag{10}$$

- $I_g$: moment of inertia of gross concrete section, neglecting reinforcement
- $I_{gM}$: moment of inertia of voided area at the minimum section (refer to Figure 13)
- $I_{gD}$: moment of inertia of voided area at the donut section (refer to Figure 13)
- $I_{cr}$: moment of inertia of cracked section transformed to concrete
- $I_e$: effective moment of inertia for computation of deflection
- $I_{test}$: moment of inertia calculated by inverse operation from test results
- $b$: width of slab
- $b'$: width of plane part of void shaper (refer to Figure 13)
- $h$: height of slab
- $h'$: height of void shaper (refer to Figure 13)
- $h''$: height of straight part of center hole (refer to Figure 13)
- $k$: factor used in calculating flexural capacity of a section
- $l$: length of clear span of specimen
- $l_a$: length between loading and support point
- $M_a$: maximum moment at the stage for which deflections are being computed

- $n$: the number of reinforcements in a layer
- $N$: the number of voids which are included in the cross-section for calculation
- $P$: load at the stage for which effective moment of inertia is computed
- $r$: radius curvature of outer side of void (refer to Figure 13)
- $r'$: radius curvature of upper part of center hole (refer to Figure 13)
- $E_c$: modulus of elasticity of concrete
- $\delta$: deflection at the stage for which effective moment of inertia is computed

Table 9 shows the comparison between the test results and theoretical value of stiffness. In all the specimens, $S_{2,Test}$ was lower than $S_{2,ACI}$ by 23~28%. It is because Equation (9) generally overestimates the effective moment of inertia of members having a lower steel ratio-like slab [16]. Based on the test results, flexural stiffness was compared between the donut-type voided slabs and solid slab. Although a gap of 20%~37% was obtained in the initial flexural stiffness before cracking ($S_{1,Test}$), it might be caused by experimental errors, because slab deflection before cracking was very small. The flexural stiffness between cracking and the yield point was lower in the donut-type voided specimens by 7.7~8.7% than the 'Solid'. It is deduced to have been caused by the reduction of the cross-section due to the voids. However, the donut-type specimens show a higher stiffness between the cracking and yield point than 'OF-V-S-R' by 7%. Therefore, it is deduced that the center hole in the void shaper improves the flexural stiffness.

**Table 9.** Moment of inertia and flexural stiffness of the specimens.

| Specimen | Theoretical Results | | | | | Test Results | | | | Comparison | | | |
|---|---|---|---|---|---|---|---|---|---|---|---|---|---|
| | $I_g$ * $\times 10^8 mm^4$ | $I_{cr}$ † $\times 10^8 mm^4$ | $I_e$ ‡ $\times 10^8 mm^4$ | $S_{1,ACI}$ § $kN/mm$ | $S_{2,ACI}$ £ $kN/mm$ | $I_{1,Test}$ € $\times 10^8 mm^4$ | $I_{2,Test}$ ¥ $\times 10^8 mm^4$ | $S_{1,Test}$ + $kN/mm$ | $S_{2,Test}$ # $kN/mm$ | $\frac{S_{1,Test}}{S_{1,ACI}}$ | $\frac{S_{2,Test}}{S_{2,ACI}}$ | $\frac{S_{1,Test}}{S_{1,Solid}}$ | $\frac{S_{2,Test}}{S_{2,Solid}}$ |
| Solid | 16.3 | 2.38 | 4.00 | 95.99 | 13.70 | 5.13 | 2.33 | 30.26 | 10.50 | 0.32 | 0.77 | 1.00 | 1.000 |
| OF-V-S-D | 15.0 (14.3) | 2.38 (2.38) | 3.53 (3.33) | 88.45 (84.53) | 12.79 (12.43) | 6.16 | 2.02 | 36.33 | 9.59 | 0.41 (0.43) | 0.75 (0.77) | 1.20 | 0.913 |
| OF-V-M-D | 15.0 (14.3) | 2.38 (2.38) | 3.53 (3.33) | 88.45 (84.53) | 12.79 (12.43) | 7.01 | 2.20 | 41.36 | 9.69 | 0.47 (0.49) | 0.76 (0.78) | 1.37 | 0.923 |
| OF-V-S-R | 14.3 | 2.38 | 3.33 | 84.53 | 12.43 | 6.22 | 1.92 | 36.66 | 8.93 | 0.43 | 0.72 | 1.21 | 0.851 |

( ): Values are calculated with the minimum cross-section area of the slab; * Moment of inertia of the gross concrete section about the centroid axis, neglecting reinforcement; † Moment of inertia of the cracked section transformed to concrete; ‡ Effective moment of inertia for the computation of deflection, which is calculated with ACI 318-08; § Secant stiffness until crack occurring ($S_{1,ACI} = P_{cr,cal}/\delta_{cr,cal}$); £ Secant stiffness from first crack to yielding point ($S_{1,ACI} = (P_{n,cal} - P_{cr,cal})/(\delta_{n,cal} - \delta_{cr,cal})$); € Moment of inertia of slab until crack occurring, which is calculated with the test results of Equation (10); ¥ Moment of inertia of slab from first crack to the yielding point, which is calculated with the test results of Equation (10); + Secant stiffness until crack occurring ($S_{1,Test} = P_{cr}/\delta_{cr}$); # Secant stiffness from first crack to the yielding point ($S_{2,Test} = (P_n - P_{cr})/(\delta_n - \delta_{cr})$).

In order to take into consideration of the influence of the reduction in the slab cross-section area, $S_{2,ACI(Void)}/S_{2,ACI(Solid)}$ calculated by using the minimum and donut section were compared with $S_{2,Test(Void)}/S_{2,Test(Solid)}$. As a result, it is found that the properties of the minimum section supply more precise and conservative estimations of deflection of donut-type voided slabs, as shown in Figure 15. Therefore, it is suggested that the minimum section should be used when the flexural stiffness of the donut-type voided slabs is estimated by the equations of current concrete structural design codes [14]. According to the previous research [16], the current design code overestimates the flexural stiffness of the members having a low reinforcement ratio, such as slabs. In addition, Al-Gasham et al. [9] argued that the reduction in the flexural stiffness of voided slabs could be attributed to the decrease in the bond strength of the rebar due to voids. No wonder the test results showed that the effective moment of inertia at the yield point was lower than the theoretical value. Therefore, further studies should be conducted to evaluate the serviceability of the donut-type voided slabs and estimate their deflection focused on the effect of reinforcement ratio and the bond characteristics.

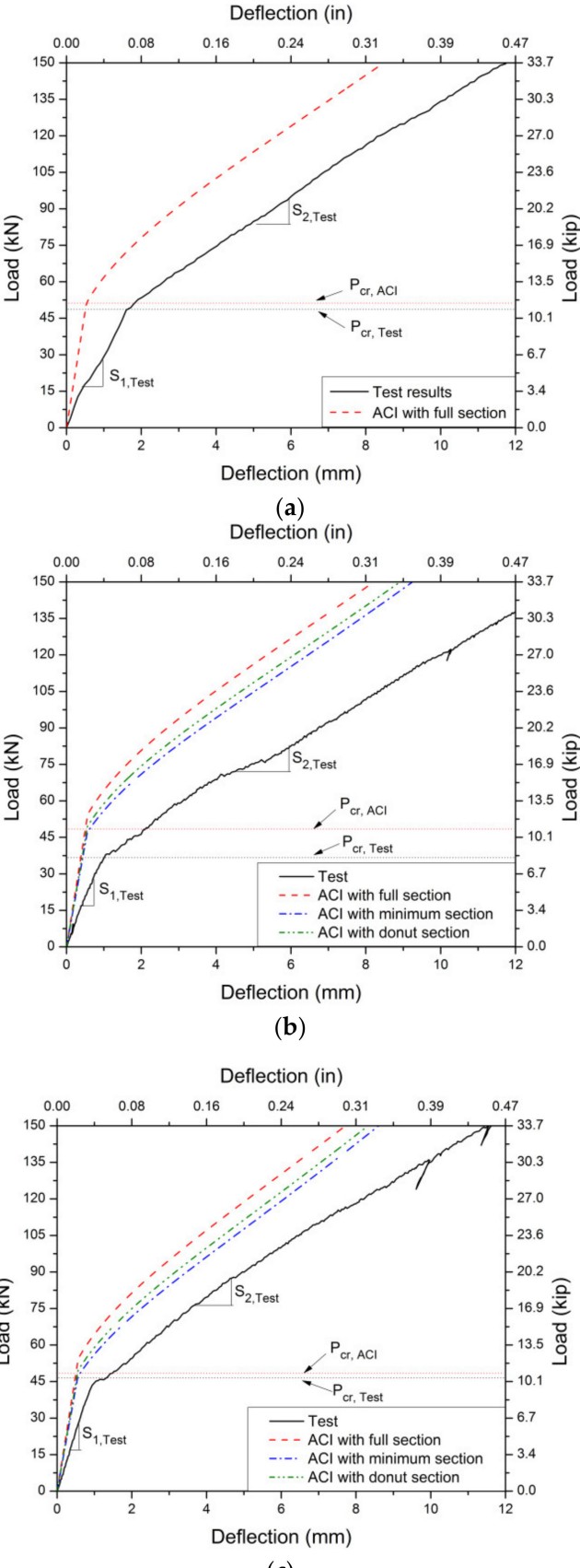

**Figure 15.** *Cont.*

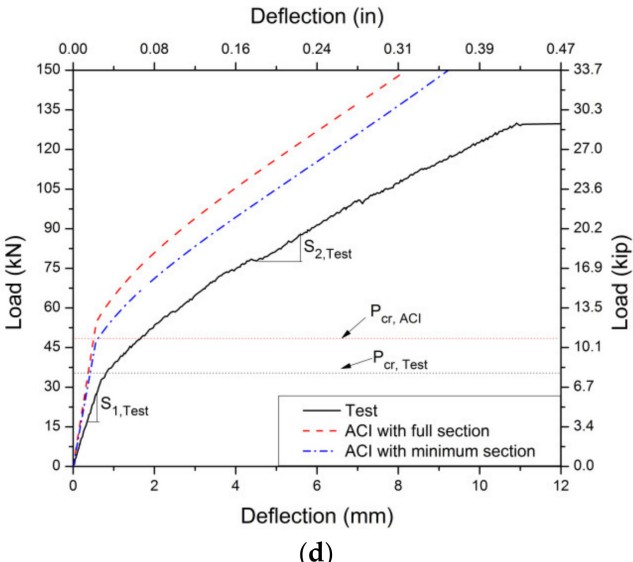

(**d**)

**Figure 15.** Comparing between theoretical and experimental stiffness based on section properties: (**a**) Solid; (**b**) OF-V-S-D; (**c**) OF-V-M-D; (**d**) OF-V-S-R.

## 5. Conclusions

In this study, voided slabs with donut-type void shapers were developed based on an analytical study in order to effectively reduce the self-weight of slabs. In addition, the flexural capacities of the donut-type voided slabs were investigated by the hole of the void shaper and the fixing method. Finally, the feasibility of applying the current design code [14] to donut-type voided slabs was examined.

1.  In terms of strength, donut-type voided slabs demonstrated higher ultimate flexural strength than the theoretical value calculated, in accordance with the current design code [14]. The test results showed the ultimate strength of 'OF-V-S-D' and 'OF-V-D-M' were 98% and 105% compared with 'Solid', and it is superior to the present voided slab systems. Therefore, donut-type voided slabs can be designed based on the nominal flexural strength prescribed in the current design code [14].

2.  In terms of ductility, the ductility of the donut-type voided slab was similar to that of the solid and superior to the present voided slabs. In addition, donut-type voided specimens were secure enough, with a displacement–ductility ratio of more than four, to perform as a general flexural member. Therefore, the donut-type voided slab can be applied as a flexural member in place of the conventional heavy solid slab.

3.  In terms of stiffness, the flexural stiffness of 'OF-V-S-D' and 'OF-V-M-D' showed 8.7% and 7.7% deterioration in flexural stiffness compared to that of 'Solid' between the initial cracking and yield point, respectively. However, donut-type voided slab specimens are similar or superior to 'OF-V-S-R' and present voided slabs. The difference in stiffness was caused by the reduction of the cross-section area by voids and the center hole of the void shaper. To take into consideration these factors, it is suggested that the minimum section should be used for the calculation of the effective moment of inertia.

4.  The test results showed the effective moment of inertia at the yield point was lower than the theoretical value. According to previous research [16], the current design code overestimates the flexural stiffness of the members having low reinforcement ratios, such as slabs. In addition, Al-Gasham et al. [9] argued that the reduction in the flexural stiffness of voided slabs could be attributed to the decrease in the bond strength of rebar due to voids. Therefore, more studies should be conducted to evaluate the serviceability of the donut-type voided slabs and estimate their deflection.

**Author Contributions:** Original draft preparation and editing, J.-H.C. and H.-K.C.; planning the test program, H.-S.J. and H.-K.C.; performing the tests and investigation, H.-S.J.; analyzing the results and reviewing the article, H.-K.C. and J.-H.C.; and supervision and review writing, H.-K.C. All authors have read and agreed to the published version of the manuscript.

**Funding:** This work was supported by the Korea Agency for Infrastructure Technology Advancement (KAIA) grant founded by the Ministry of LAND, Infrastructure and Transport (22CTAP-C164325-02).

**Institutional Review Board Statement:** Not applicable.

**Informed Consent Statement:** Not applicable.

**Data Availability Statement:** Data are contained within the article.

**Conflicts of Interest:** The authors declare no conflict of interest.

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
