# Peer review of "Flexural Strength and Stiffness of Donut-Type Voided Slab"

_applsci, doi:10.3390/app12125782_

Round 1

Reviewer 1 Report

The manuscript compares the flexural strength capacity of donut type hollow slabs and solid slabs. The authors have compared different geometries for the hollow shapers for the slabs with a slab of solid geometry using finite element methods and experimental data. The manuscript is very relevant to the current research needs. However, a few comments need to be addressed before the manuscript can be accepted for publication.

Introduction section - Please carry out a comprehensive literature review for the problem at hand, identify issues with the current studies, and present what your hypothesis and experimental objectives are. At the moment, the introduction section does not say much except that two companies developed a slab system, which a couple of studies showed that they have lower flexural capacities.

Table 1 - The authors may please comment on how much decrease is acceptable. While, yes, it is good to have a similar strength and stiffness to that of a solid slab, is it acceptable to have a decrease of up to 15% without affecting the serviceability conditions?

Fig 11 - It appears that the theoretical calculations always overestimate the stiffness values. I would comment on addressing this in the theoretical calculations by way of correction factors/equations in the discussion section.

I am curious as to how much materials savings is achieved with each configuration of the donut holes in the hollow slabs. It would be great to see something like a plot on materials savings/volume empty versus property improvement for choosing an "optimal" section.

Author Response

I appreciate your thoughtful comments and suggestions on my work.

Reviewer 2 Report

Review report on

FLEXURAL STRENGTH AND STIFFNESS OF DONUT TYPE BIAXIAL HOLLOW SLAB  

In the current manuscript, the flexural capacities of donut type biaxial hollow slab was presented. Based on FE analysis, donut type hollow-shaper was suggested as the optimal shape. Flexural tests were conducted for 4 specimens to verify flexural capacities of that. The one was solid slab and others were biaxial hollow slabs. Donut type hollow slabs had different fixing methods of hollow-shapers. The one was used diagonal steel-cage and the other was fixed without additional devices. The rest specimen was non-donut type hollow slab. Test results showed that flexural strength of donut type hollow slabs were equivalent to 98% and 105% of solid slab. It also showed enough displacement-ductility ratio more than 4. The stiffness of this hollow slab was decreased about 8~9% comparing with the solid slab. However, it was improved up to 7% compared to non-donut type hollow slab.  

The paper is well-organized and includes new contributions with good merits for publication. I humbly ask the authors to carefully read the attached concerns and make major modifications to enhance the presentation of their paper.

- The authors must explicitly declare the assumptions and limitations of their model.  

- What is the main objective behind the current study? It is beneficial for the readers to add more explanations about the novel contribution of this method from theoretical/experimental viewpoints.

- It is recommended to present the real applications of the presented model/method in industry or different scientific fields with their schematic configurations.

- Some parameters need to be defined just after appearing in the text.

- The introduction part needs to be extended by discussing more relevant papers. The authors should appropriately extend this section by discussing more relevant works focusing on different methods and models in the topic of concrete performance and analysis. For example, it is suggested to read and discuss about the following relevant works:

- Abbès, F., Abbès, B., Benkabou, R., Asroun, A. A FEM Multiscale Homogenization Procedure using Nanoindentation for High Performance Concrete. Journal of Applied and Computational Mechanics, 2020; 6(3): 493-504.

and other related works.

- It is suggested to add more in-depth explanation of the model, its justification and more discussions on the results.

- The paper should be carefully double-checked from grammatical point of view.

- The resolution of presented figures are poor and need strict modification for quality publication.

Author Response

(The authors gave the same response as above.)

Reviewer 3 Report

The manuscript is on FLEXURAL STRENGTH AND STIFFNESS OF DONUT TYPE BIAXIAL HOLLOW SLA. The manuscript requires the following revision before it can be considered for publication

1. Abstract - the abstract should have the following elements in a paragraph: (1) problem to be solved with objective (2) methodology adopted (3) key findings and (4) practical implications on key findings

2. There should be a section on methods and materials. here there should be a section on design of experiment. 

3. The manuscript talks about optimization, however the argument on the objective function and related constraints are not very clear from the write up. 

4. The authors need to specify the scope and limitation of the findings, academic and practical terms

5. All equations neds to be explained in the manuscript, state reference where required and define the terms used etc.

6. The research gap that support the aim of the study needs to be enhanced further with latest and more comprehensive review

7.In table 2, can the authors explain further the rationale for selecting the shapes

8. From the writeup, it is not clear if the finite element model was validated and if any mesh sensitivity analysis was done?

9. based on Figure 1, in the elastic zone, the performance of all the designs are not really significant. What is the practical motivation of such findings?

Author Response

(The authors gave the same response as above.)

Round 2

Reviewer 2 Report

Accept as is.

Reviewer 3 Report

Can be accepted once converted to journal format